# Compatibility of Dual-Cure Core Materials with Self-Etching Adhesives

**DOI:** 10.3390/dj13070276

**Published:** 2025-06-20

**Authors:** Zachary K. Greene, Augusto A. Robles, Nathaniel C. Lawson

**Affiliations:** 1Division of Biomaterials, UAB School of Dentistry, Birmingham, AL 35233, USA; zkgreene@uab.edu; 2Division of General Dentistry, UAB School of Dentistry, Birmingham, AL 35233, USA; arobles@uab.edu

**Keywords:** shear bond strength, core buildup, universal adhesive, self-etch adhesive

## Abstract

**Background/Objectives**: A material incompatibility has been established between self-etching adhesives and amine-containing dual-cure resin composite materials used for core buildups. This study aims to compare the dentin bond strength of several amine-containing and amine-free core materials using self-etching adhesives with different pHs. **Methods**: Extracted human molars were mounted in acrylic and ground flat with 320-grit silicon carbide paper. Next, 520 specimens (n = 10/group) were assigned to a dual-cure core buildup material group (10 amine-containing, 2 amine-free, and 1 reference light-cure only bulk fill flowable composite) and assigned to a self-etching adhesive subgroup (pH levels of approximately 1.0, 3.0, and 4.0). Within 4 h of surface preparation, the adhesive corresponding to the specimen’s subgroup was applied and light-cured. Composite buttons for the assigned dual-cure core material of each group were placed using a bonding clamp apparatus, allowed to self-cure for 2 h at 37 °C, and then unclamped. An additional group with one adhesive (pH = 3.0) was prepared in which the dual-cure core materials were light-cured. The bonded specimens were stored in water at 37 °C for 24 h. The specimens were mounted on a testing clamp and de-bonded in a universal testing machine with a load applied to a circular notched-edge blade at a crosshead speed of 1 mm/min until bond failure. The maximum load divided by the area of the button was recorded as the shear bond strength. The data was analyzed via 2-way ANOVA. **Results**: The analysis of bond strength via 2-way ANOVA determined statistically significant differences between the adhesives, the core materials, and their interaction (*p* < 0.01). There was a general trend in shear bond strength for the adhesives, where pH 4.0 > 3.0 > 1.0. The amine-free core materials consistently demonstrated higher shear bond strengths as compared to the other core materials when chemically cured only. Light-curing improved bond strength for some materials with perceived incompatibility. **Conclusions**: The results of this study suggest that an incompatibility can exist between self-etching adhesives and dual-cure resin composite core materials. A decrease in the pH of the utilized adhesive corresponded to a decrease in the bond strength of dual-cure core materials when self-curing. This incompatibility may be minimized with the use of core materials formulated with amine-free chemistry. Alternatively, the dual-cure core materials may be light-cured.

## 1. Introduction

The placement of a core buildup is essential for restoring structurally compromised and endodontically treated teeth [1,2,3]. A key feature of materials used for a core buildup is their ability to fully polymerize, even when placed in bulk and without access to light polymerization. Therefore, dual-cure resin composites are often used for core buildups. Many modern dual-cure resin composites achieve self-curing via redox-initiated polymerization between a peroxide and a tertiary amine. Mixing an accelerator paste (with the amine) and an initiator paste (with a peroxide such as benzoyl peroxide) triggers the polymerization of the monomers and oligomers [4].

The bond of dual-cure core materials to tooth structure is critical, especially in cases involving extensive tooth structure loss (e.g., missing cusps) or limited retention and resistance forms (e.g., following crown preparation). Adhesion between tooth structure and core materials is achieved through adhesives. The introduction of simplified, self-etching adhesive systems in the 2000s and 2010s (i.e., sixth-generation, seventh-generation, and universal adhesives) reduced clinical steps and technique-sensitivity relative to previous etch-and-rinse adhesives [5,6,7]. However, bond compatibility issues were noted between simplified self-etch adhesives and dual- or self-cure resin composites [8,9,10,11,12,13,14,15,16,17,18,19,20,21,22]. Bond incompatibility implies a lower bond of self-cure materials with self-etch compared to etch-and-rinse adhesives or a lower bond of self-cure compared to light-cure materials when using self-etch adhesives. Earlier bonding systems (fourth- and fifth-generation adhesives) relied on etch-and-rinse protocols, which used phosphoric acid to remove the remnant smear layer and demineralize tooth structure; therefore, the primers and bonding resins of these systems were not acidic. In contrast, newer self-etching adhesives (sixth-generation, seventh-generation, and universal adhesives) use self-contained acidic primers to modify the smear layer and partially demineralize tooth structure without rinsing [5,6]. Sixth-generation adhesives (two-step self-etch) employ separate bonding resins free of acidic monomers that cover the initially applied acidic primer, forming a hydrophobic and pH-neutral bonding surface [1]. In contrast, seventh-generation and universal systems (one-step self-etch) leave a thin, uncured, acidic, and hydrophilic layer in direct contact with the overlying restorative composite [1]. This residual acidic layer is believed to interfere with the polymerization of dual- and self-cure composites. Specifically, acidic monomers in the oxygen inhibition layer may react with tertiary amine co-initiators, preventing the redox reaction between the amine and the peroxide [23]. This results in incomplete polymerization at the adhesive–composite interface, leading to a decreased bond [8,9,10,11,12,13,14,15,16,17,18,19,20,21,22].

To address compatibility issues with dual-cure composites, manufacturers have developed dual-cure activators for use with self-etching universal adhesives. These liquid co-initiators, typically sodium salts of aromatic sulphinic acid, are mixed with the adhesive prior to application. The sulphinic acid reacts with acidic monomers, and this so-called side reaction aims to prevent the inactivation of the tertiary amine [23]. Unfortunately, previous studies have reported that the use of a dual-cure activator with self-etching adhesives did not improve their bond strength when used with a dual-cure composite in a self-cure mode [15,24]. Dual-cure activators used with dual-cure composites in a light-cure mode were only beneficial for certain adhesives and certain composites [25,26,27], and actually reduced dentin bond strength for some materials [28]. The reduction in bond was attributed to a dilution of the adhesive [28].

Due to the inconsistency of the success of dual-cure activators, another strategy to address the bond incompatibility is the use of dual-cure materials with alternate chemical-cure mechanisms. One approach is the use of amine-free dual-cure materials. Hydroperoxides and thioureas such as cumene hydroperoxide (CHP) and 1-(2-pyridyl)-2-thiourea (PTU) are novel co-initiators, in which the hydroperoxide is the oxidizing agent and the thiourea is the reducing agent. Additionally, a metal salt such as a copper II salt serves as a catalyst for this novel polymerization reaction, and alterations in the amount of the metal salt present can change the speed at which polymerization occurs [29,30,31]. This initiator system is used in several commercially available materials, including Bulk EZ Plus (Zest Dental Solutions, Carlsbad, CA, USA) and Evanesce Bulk Fill (Clinician’s Choice, Lompoc, CA, USA).

Some dual-cure core materials incorporate resin-modified glass ionomer components, such as Activa BioActive Restorative (Pulpdent Corporation, Watertown, MA, USA) and Geristore (DenMat, Lompoc, CA, USA). Traditional glass ionomer cements consist of fluoroaluminosilicate glass particles, polyacrylic acid, and water. When mixed, an acid–base reaction occurs: basic groups in the glass powder react with the hydrated protons in the polyacrylic acid, releasing sodium and calcium ions, followed by aluminum ions [32]. These ions form crosslinks with the polyacid chains, creating an insoluble polysalt matrix that solidifies the material [32]. Glass ionomer cements are notable for their ability to chemically bond to tooth structure through ionic interactions. To address the limitations in the mechanical properties of glass ionomers, water-soluble methacrylate monomers and photoinitiators were added to form resin-modified glass ionomers. These modifications enable both light- and self-curing, making them true dual-cure materials [33].

The aim of this in vitro study was to examine the shear bond strength to the dentin of amine-containing, amine-free, and acid–base-containing dual-cure resin composite core materials (in their chemical-cure mode) using three different self-etching adhesives (OptiBond Universal, Scotchbond Universal Plus, and Clearfil SE Bond 2) with different pH values. It was postulated that (1) amine-free and acid–base-containing resin composite core materials would achieve stronger bonds than amine-containing materials in their chemical cure mode and (2) the increased acidity of simplified, self-etching universal adhesives would be more detrimental to the bond strength of dual-cure core materials than a two-step self-etching adhesive. Using one of the adhesives (Scotchbond Universal Plus, Solventum, St. Paul, MN, USA), the shear bond strength of the core materials was additionally examined when the core materials were light-cured. This group tested the theory whether decreased bonding with acidic adhesives is in fact caused by alterations in the chemical-cure mechanism of the core material. The null hypotheses were that there would be no difference in the bond strength between the (1) core materials and (2) adhesives tested.

## 2. Materials and Methods

Approximately 300 freshly extracted, caries-free human molars and premolars were collected and stored in a 1:10 formalternate-to-water solution (Flinn Scientific, Batavia, IL, USA) at 4 °C in a refrigerator for up to six months after the estimated date of extraction (IRB-300014120). Formalternate, a formaldehyde alternative, was used to minimize microbial growth. Teeth with discoloration, fractures, or a history of restorative treatment were excluded from the study. After examination, 260 acceptable teeth were selected. Each tooth was sectioned in half, yielding a total of 520 specimens. These specimens were then divided into 52 groups (n = 10 per group) for a 13 × 4 study design.

Specimens were prepared for shear bond strength testing following the notched-edge shear bond strength protocol (ISO 29022:2013E) [34]. Using a sectioning saw (IsoMet, Buehler, Lake Bluff, IL, USA), the teeth were sectioned in half along either the central groove or buccal groove to obtain two caries-free halves. The sectioned halves were placed into cylindrical plastic molds with the enamel buccal, mesial, distal, or lingual surface facing down (whichever had the largest surface area). The sectioned teeth were embedded in acrylic blocks using auto-polymerizing cross-linked acrylic (Yates Motloid, Elmhurst, IL, USA). Once the acrylic set, a flat bonding surface was created with a diamond cutting disk attached to a fixture in a model trimmer designed to ensure a perpendicular surface. The specimens were ground until enough superficial dentin was exposed to allow for dentin bonding. The external dentin surface was wet-polished to 320-grit SiC paper using a polishing wheel. The specimens were divided into 13 groups to be used to test 12 different dual-cure core materials and a reference light-cure flowable resin composite (Table 1). Each of the 13 groups were subsequently divided into 4 subgroups (n = 10) and received treatment with one of the 3 examined adhesives (Table 2); subgroups 1 through 3 were designated for core self-curing, while subgroup 4 was designated for light-curing of the dual-cure core materials. The three examined adhesives were chosen based on their pH values. The approximate pH of the adhesives was tested by rubbing each adhesive on a pH testing strip (MQuant pH indicator strips, testing pH range of 0–6; Milipore Sigma, Burlington, MA, USA).

The specimens were surface-treated with the adhesive corresponding to their assigned subgroups according to the manufacturers’ instructions for use (Table 2). An Elipar S10 curing light (Solventum, St. Paul, MN, USA) with an irradiance of at least 1000 mW/cm^2^ was used to cure the adhesives per manufacturers’ recommended curing times. The specimens were then placed into the Ultradent shear bond strength bonding clamp apparatus (Ultradent, South Jordan, UT, USA). The plastic mold insert was lowered onto the specimen (Figure 1, left), ensuring that dentin was exposed. After securing the specimen, the dual-cure core material corresponding to the specimen’s assigned group was injected into the white plastic mold insert, which formed a 2.38 mm diameter button.

While secured in the Ultradent bonding clamps, the specimens were placed in an incubator at 37 °C for 2 h to allow for chemical-curing. The choice to allow 2 h of chemical-cure prior to separating the mold was based on pilot testing in which specimens failed prematurely if separated earlier. All specimens tested with the reference light-cure flowable resin composite were light-cured according to manufacturer’s instructions. A fourth subgroup (n = 10) was tested for the 12 dual-cure materials in which Scotchbond Universal Plus was applied and light-cured per manufacturer’s instructions for use, and the core materials were light-cured according to each manufacturer’s instructions. After preparation, all specimens were stored in distilled water in an incubator at 37 °C for 24 h.

The specimens were mounted into a metal testing clamp (Figure 1, center). The testing clamp was then placed into a custom steel fixture which is designed to resist lateral forces during load application. The custom steel fixture contains a circular notched-edge blade for applying load to the button (Figure 1, right). A load was applied to the custom fixture in compression using a universal testing machine (Instron 5565, 500N load cell, Canton, MA, USA) at a crosshead speed of 1 mm/min until bond failure occurred. The maximum load applied at the time of failure was divided by the surface area of the button (4.45 mm^2^) and recorded as the shear bond strength. A 2-way ANOVA was performed for shear bond strength for the factors core material and adhesive system using a statistical software program (IBM SPSS Statistics, v29, IBM Corp, Armonk, NY, USA) (α = 0.05 for all analyses). Samples were analyzed for failure mode using a 3D measurement system (VR-5000, Keyence, Osaka, Japan) to determine if bond failures were adhesive or cohesive. The use of 3D measurement allowed for an examination of the plane of debonding (Figure 2). As no specimens were purely cohesive, a cohesive failure infers that there was some portion of the bonding area that failed in cohesion.

## 3. Results

A post hoc power analysis was conducted using G*Power (version 3) to determine the achieved power for a two-way ANOVA with a 13 × 4 factorial design. Given an effect size of f = 0.648 (partial η^2^ = 0.296), an α level of 0.05, and a total sample size of 520, the analysis indicated that the achieved power (1 − β) was 1.00. This suggests that the study was sufficiently powered to detect the observed effect.

The mean and standard deviation of shear bond strength are presented in Table 3 and Figure 3. A two-way ANOVA was conducted to examine the effects of core material and adhesive system on bond strength. Normality was assessed by using the Shapiro–Wilk normality test, and the homogeneity of variances was confirmed by the Levene test. The ANOVA table is presented in Table 4. A statistically significant interaction was found between core material and adhesive system. An analysis of simple main effects for core material and adhesive system was performed with statistical significance receiving a Bonferroni adjustment. The results of the pairwise comparisons are presented in Table 3.

The results of the failure mode analysis are summarized in Table 5. The majority of failures were adhesive in nature. These failures were characterized by smooth surfaces with no remnants of core material and no visible loss of tooth structure. It was not possible to determine whether the adhesive failures occurred at the dentin–adhesive interface or at the adhesive–core material interface. Among the cohesive failures, most involved a loss of dentin and were associated with higher shear bond strength values.

## 4. Discussion

The first null hypothesis that there would be no difference in the shear bond strength between different materials was rejected. The results of the two-way ANOVA indicated that there were significant differences between materials, and the interaction between core materials and adhesive systems was also significant. Therefore, the differences in bond between materials is dependent on which adhesive was used. With the use of OptiBond Universal, materials can be grouped into different clusters. Filtek Bulk Fill Flowable, Evanesce Bulk Fill, and Active BioActive Restorative outperformed a group of materials with very low bond strength, including CompCore AF, CorePaste XP, LuxaCore Z Dual, and Build-It FR. Geristore and Bulk EZ Plus outperformed two of those low performing materials. The other materials were intermediate. With Scotchbond Universal Plus, the materials Evanesce Bulk Fill, Filtek Bulk Fill Flowable, Activa BioActive Restorative, and Bulk EZ Plus outperformed the two materials with the lowest bond strength, LuxaCore Z Dual and CompCore AF.

A summary of these results reveals that Active BioActive Restorative, Bulk EZ Plus, and Evanesce Bulk Fill were consistently high-performing dual-cure materials when the acidic single-bottle adhesives (OptiBond Universal and Scotchbond Universal Plus) were used. These results support the postulation that amine-free and acid–base-containing dual-cure materials could overcome the incompatibility with self-etch adhesives. Bulk EZ Plus and Evanesce Bulk Fill are both amine-free and use cumene hydroperoxide and thioureas as co-initiators. The positive results with Activa BioActive Restorative may be in part credited to the acid–base (i.e., glass ionomer cement) reaction occurring within the material.

The very low bond strength (less than 10 MPa) of many of the materials when used with OptiBond Universal demonstrates the incompatibility of many dual-cure materials with acidic, simplified self-etching adhesives. OptiBond Universal is recommended to be mixed with a dual-cure activator when paired with a dual- or self-cure core material; however, the dual-cure activator was not used in the current study. The dual-cure activator was excluded from this group to scientifically assess the intrinsic incompatibility of a low-pH adhesive with dual-cure materials. Previous studies have demonstrated that dual-cure activators do not improve bond strength when used with a dual-cure composite in a self-cure mode [15,24]. Additionally, the omission of the dual-cure activator simulated clinical scenarios in which the activator may be omitted, whether due to oversight, a lack of availability, or a lack of knowledge.

When Clearfil SE Bond 2 was used as the bonding agent, the dual-cure materials Bulk EZ Plus and Evanesce Bulk Fill still produced a significantly higher bond strength than most other materials. When Scotchbond Universal Plus was used as the adhesive and the core materials were light-cured, Bulk EZ Plus and Evanesce Bulk Fill no longer outperformed the other materials. These results suggest that the cumene hydroperoxide and thiourea co-initiators used in Bulk EZ Plus and Evanesce Bulk Fill may be more efficient in developing an adhesive bond via chemical-curing than amine-based co-initiators, regardless of the acidity of the adhesive.

Results of the failure mode analysis reveal that most of the specimens of the amine-containing materials failed in an adhesive mode. This observation was expected due to the inactivation of the tertiary amine co-initiators by the acidity of the self-etch adhesives. There were a higher proportion of cohesive failures in groups with higher bond strength values, such as the amine-free and acid–base-containing materials. Also, there were more cohesive failures noted when the core materials were light-cured, a trend also reported in previous studies [15,24].

The second null hypothesis that the adhesive system would not affect the bond strength of the different core materials was rejected. The results of the two-way ANOVA indicated that there were significant differences between adhesive systems and their interaction with different core materials. For eight of the materials tested, a higher bond strength was achieved with Scotchbond Universal Plus than OptiBond Universal, and almost every material achieved a higher bond strength with Clearfil SE Bond 2 than OptiBond Universal. For four materials, a higher bond was achieved with Clearfil SE Bond 2 than Scotchbond Universal Plus. The adhesives rank, in increasing acidity, from Clearfil SE Bond 2 (pH = 4.0 to 4.5) to Scotchbond Universal Plus (pH = 3.0 to 3.5) to OptiBond Universal (pH = 1.0 to 1.5). These results demonstrate a trend that a higher bond can be achieved with dual-cure core materials as the pH of the adhesive is increased. This trend has been described previously [12,14].

The pH of Scotchbond Universal Plus was reported by the manufacturer to be around 2.7 and was approximated to be between 3.0 and 3.5 using pH strips [35]. In a previous study, adhesives with pH values between 1.0 and 2.8 were found to completely inhibit the chemical cure of amine-based dual-cure materials. However, when these adhesives were mixed with dual-cure activators that raised the pH to a range of 3.0 to 3.5, a proper polymerization of the dual-cure materials was achieved [23]. Therefore, the pH of Scotchbond Universal Plus falls within a critical transitional range—high enough to permit some degree of chemical-curing, but still low enough to risk incompatibility. Scotchbond Universal Plus is also reported to contain a built-in dual-cure activator, which may contribute to its superior bond strength compared to OptiBond Universal.

The behavior of the reference light-cure only composite (Filtek Bulk Fill Flowable) demonstrated a higher bond with Clearfil SE Bond 2 than the other two adhesives. These results suggest that the superior performance of Clearfil SE Bond 2 with the dual-cure materials is not just credited to its pH, but also the mechanism by which it achieves adhesion. A previous study has also reported a higher bond strength with Clearfil SE Bond 2 than Scotchbond Universal [7].

All of the amine-containing dual-cure core materials except two achieved a statistically higher bond strength with Scotchbond Universal Plus when the core material was light-cured. This observation was reported previously [12]. These results either provide evidence that the acidity of the adhesive interfered with the chemical cure of the core material, or that a higher bond could be achieved when the core material was light-cured rather than chemically cured. Light-curing dual-cure materials achieve a greater degree of conversion than allowing them to chemically cure alone [36]. A previous study reported significantly lower shear bond values for similar combinations of materials as used in this study (Scotchbond Universal with Core Paste XP at 6.5 MPa and CompCore AF at 17.2 MPa when the core materials were light-cured) [26]. In that study, the Scotchbond Universal was used with a dual-cure activator, which has been shown to reduce bond strength in some instances [28].

There are several limitations of the current study. First, there was a high level of variability in the data that could be due to the natural variability in the microstructure of extracted teeth as well as technique sensitivity when preparing specimens with delicate bonds. Increasing sample size to decrease data variability was limited due to the difficulty in obtaining freshly extracted teeth. Second, only one adhesive was evaluated when the dual-cure core material was light-cured. This limitation was due to the difficulty in obtaining the required number of freshly extracted teeth required to test the additional groups. Third, the specimens were not thermocycled or aged. Thermocycling would simulate the differential contraction of materials at the bond interface and aging would allow hydrolytic or enzymatic bond degradation. These factors may have led to an even poorer performance of material combinations with bond incompatibility. Regardless, the bond incompatibility between acidic bonding agents and dual-cure materials was demonstrated in the 24-h bond strength results presented in this study. Future studies may examine the effects of adding a dual-cure activator to simplified self-etching adhesives on the shear bond strengths of contemporary dual-cure materials.

## 5. Conclusions

Several conclusions may be drawn from this study. First, an incompatibility exists between many amine-containing dual-cure core materials and very low-pH (1.0 to 1.5) adhesives. Second, a less acidic (pH = 3.0 to 3.5) adhesive with a self-contained dual-cure activator demonstrated greater compatibility with dual-cure core materials than the very low-pH adhesive. Third, dual-cure core materials like Bulk EZ Plus and Evanesce Bulk Fill, which use amine-free co-initiators (cumene hydroperoxide and thioureas) and one of the acid–base-containing dual-cure core materials (Active BioActive Restorative) demonstrated consistently high bond strengths with acidic adhesives, minimizing compatibility issues. Fifth, light-curing core materials generally led to higher bond strength, particularly for amine-based dual-cure materials, indicating that light activation can partially overcome chemical incompatibility with acidic adhesives.

## Figures and Tables

**Figure 1 dentistry-13-00276-f001:**
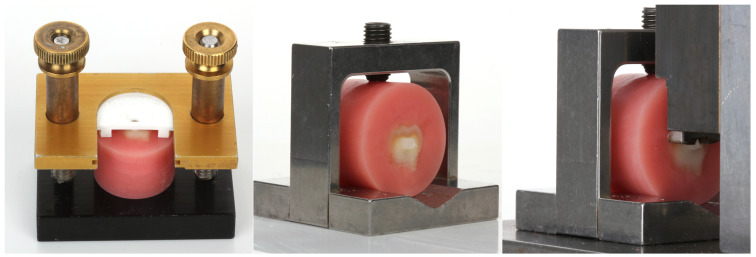
Ultradent shear bond strength apparatus (**left**); Button of core material bonded to the tooth specimen (**center**); Circular notched-edge blade for applying load to the button (**right**).

**Figure 2 dentistry-13-00276-f002:**
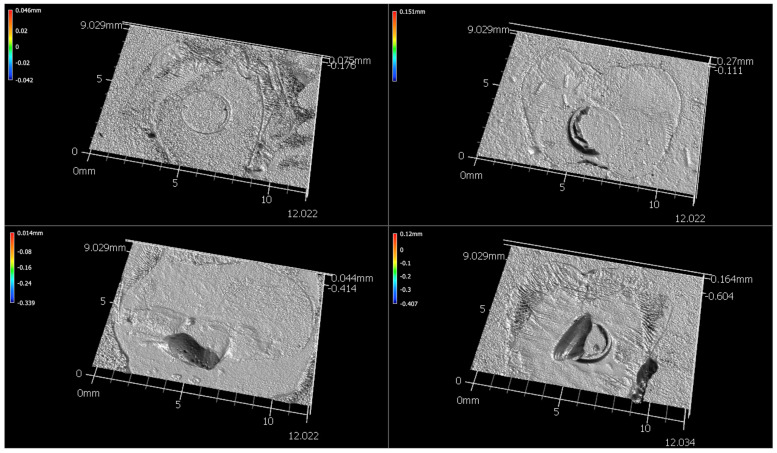
Example failure modes: Adhesive (**top left**), Cohesive within tooth (**bottom left**), Cohesive within core materials (**top left**), Cohesive within core and tooth (**bottom right**).

**Figure 3 dentistry-13-00276-f003:**
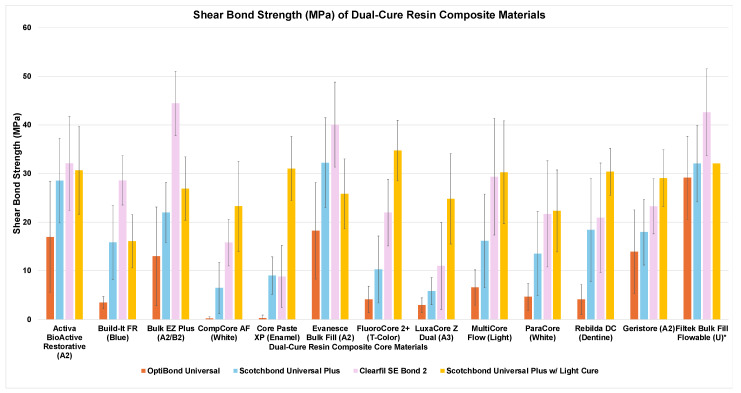
Shear bond strength (average ± standard deviation) of dual-cure resin composite core materials using different adhesive systems. * Light-cure only control group in which core was light-cured for all groups.

**Table 1 dentistry-13-00276-t001:** Core materials used in this study.

Material Manufacturer LOT	Material Type	Relevant Material Composition
Activa BioActive-Restorative Pulpdent Corp Watertown, MA, USA LOT: 240129	acid–base-containing	UDMA and other methacrylates; Modified polyacrylic acid; Silica; Sodium fluoride
Build-It FR Pentron (Kerr) Orange, CA, USA LOT: 9604454	amine-containing	Bis-GMA, UDMA, HDDMA, Silane-treated barium boro-alumino silicate glass, Silane-treated chopped glass fibers, Pigments, UV absorbers, Initiators
Bulk EZ Plus Zest Dental Solutions Carlsbad, CA, USA LOT: L38KK	amine-free	Zirconia silica filler, Radiopaque filler, Ethoxylated Bis-DMA, TEGDMA, Bis-GMA, UDMA, Initiators (CHP, PTU, copper (II) catalyst)
CompCore AF Premier Dental Plymouth Meeting, PA, USA LOT: 5267-20QCW	amine-containing	Catalyst: TEGDMA; Benzoyl peroxide; Fumed silica Base: TEGDMA; Co-initiator; Photoinitiator; Fumed silica
Core Paste XP Syringe Den-Mat Lompoc, CA, USA LOT: 2419200010	amine-containing	Ethoxylated Bis-DMA; TEGDMA; Silica; DHEPT; 2-(2′ hydroxy-5′-octylphenyl) benzotriazole; Benzoyl peroxide; TPO; BHT; Ethyl 4-dimethylaminobenzoate; Camphorquinone
Evanesce Bulk Fill Clinician’s Choice Lompoc, CA, USA LOT: 150CT24	amine-free	Glass filler, TEGDMA, Titanium dioxide, CHP, BHT, Copper (II) catalyst, Iron oxide.
FluoroCore 2+ Dentsply Sirona Charlotte, NC, USA LOT: 00150296	amine-containing	Base Paste: Silane-treated barium boro-alumino silicate glass; Fumed silica; Aluminum oxide; UDMA; Urethane modified Bis-GMA; Polymerizable methacrylate resins Catalyst Paste: Silane-treated barium boro-alumino silicate glass; Fumed silica; Aluminum oxide; Bis-GMA; Polymerizable dimethacrylate resin; Benzoyl peroxide
Geristore Den-Mat Lompoc, CA, USA LOT: 2408700069	acid–base-containing	Benzoyl peroxide; Ethyl 4-dimethylaminobenzoate; Ethoxylated Bis-DMA; Polyacrylic acid; HEMA; NTG-GMA; Bis[2-[(2-methyl-1-oxoallyl)oxy]ethyl] dihydrogen; Tartaric acid; Camphorquinone; Silica; 2-(2′ hydroxy-5′-octylphenyl) benzotriazole; BHT
LuxaCore Z Dual DMG America Ridgefield Park, NJ, USA LOT: 296096	amine-containing	Silica; Bis-GMA; UDMA; HDDMA; DDDMA; MMHE; EHDAB; Benzoyl peroxide
Multicore Flow Ivoclar Vivadent Schaan, Liechtenstein LOT: Z06NHR	amine-containing	Base: Ytterbium trifluoride; Bis-GMA; UDMA; TEGDMA Catalyst: Ytterbium trifluoride; Bis-GMA; UDMA; TEGDMA; Dibenzoyl peroxide
ParaCore Coltene Altstätten, Switzerland LOT: M66296	amine-containing	UDMA; TMPTMA; Bis-GMA; TEGDMA; Dibenzoyl peroxide; Benzoyl peroxide; Sodium fluoride; Barium glass; Amorphous silica
Rebilda DC Voco GmbH Cuxhaven, Germany LOT: 2424665	amine-containing	BHT; Amines; Benzoyl peroxide; Barium aluminum borosilicate glass; UDMA; HEDMA; DDDMA; Pyrogenic silica; Bis-GMA; Initiators; Stabilizers; Pigments.
Filtek Bulk Fill Flowable Universal Solventum St. Paul, MN, USA LOT: 10107433	Light-cured bulk-fill composite	Silane-treated ceramic; UDMA; Substituted dimethacrylate; Ytterbium fluoride; Bis-GMA; Bis-EMA-6; TEGDMA; EDMAB

**Table 2 dentistry-13-00276-t002:** Adhesive materials used in this study.

Material Manufacturer LOT	pH (Manufacturer Reported)	pH (Measured with Strips)	Bonding Protocol
OptiBond Universal Kerr Orange, CA, USA LOT: A172205	Not reported	1.0–1.5	20 s application, 5 s air thin, 5 s cure
Scotchbond Universal Plus Solventum St. Paul, MN, USA LOT: 10513692	2.7^30^	3.0–3.5	20 s application, 5 s air thin, 10 s cure
Clearfil SE Bond 2 Kuraray Noritake Tokyo, Japan LOT: 3E0323 (Control)	Not reported	4.0–4.5 (2nd bottle)	(1st bottle) 20 s application, air thin (2nd bottle) 20 s application, air thin, 10 s cure

**Table 3 dentistry-13-00276-t003:** Shear bond strength (MPa, mean ± standard deviation) of different core materials using different adhesive systems.

	OptiBond Universal	Scotchbond Universal Plus	Clearfil SE Bond 2	Scotchbond Universal Plus (Core Light-Cured)
Activa BioActive Restorative	17.0 ± 11.4 CD,a	28.5 ± 8.7 DE,b	32.1 ± 9.7 CDE,b	30.6 ± 9.0 BC,b
Build-It FR	3.4 ± 1.3 AB,a	15.8 ± 7.6 ABC,b	28.6 ± 5.1 CD,c	16.1 ± 5.5 A,b
Bulk EZ Plus	13.0 ± 10.1 BCD,a	22.0 ± 6.2 CD,b	44.5 ± 6.6 E,c	26.9 ± 6.5 ABC,b
CompCore AF	0.2 ± 0.4 A,a	6.4 ± 5.2 AB,b	15.8 ± 4.8 AB,bc	23.3 ± 9.3 ABC,c
Core Paste XP Syringe	0.3 ± 0.6 A,a	9.0 ± 3.8 ABC,a	8.8 ± 6.4 A,a	31.0 ± 6.6 BC,b
Evanesce Bulk Fill	18.2 ± 9.9 DE,a	32.2 ± 9.2 E,b	40.1 ± 8.7 DE,b	27.8 ± 4.3 ABC,b
FluoroCore 2+	4.1 ± 2.7 ABC,a	10.2 ± 6.9 ABC,a	21.9 ± 6.8 ABC,b	34.7 ± 6.2 C,c
Geristore	13.9 ± 8.6 BCD,a	17.9 ± 6.8 BCD,ab	23.3 ± 5.7 BC,bc	29.0 ± 5.9 BC,c
LuxaCore Z Dual	2.9 ± 1.5 AB,a	5.8 ± 2.8 A,a	11.0 ± 8.9 A,a	24.8 ± 9.3 ABC,b
Multicore Flow	6.6 ± 3.6 ABCD,a	16.1 ± 9.6 ABCD,a	29.3 ± 12 CD,b	30.2 ± 10.6 BC,b
ParaCore	4.7 ± 2.8 ABC,a	13.5 ± 8.7 ABC,ab	21.7 ± 10.9 ABC,b	22.3 ± 8.4 BC,b
Rebilda DC	4.1 ± 3.1 ABC,a	18.4 ± 10.6 BCD,b	20.9 ± 11.3 ABC,b	30.4 ± 4.8 BC,c
Filtek Bulk Fill Flowable Universal *	29.1 ± 8.5 E,a	32.0 ± 7.9 E,a	42.6 ± 8.9 E,b	NA

Significant differences between materials are indicated with different uppercase letters in each column and between adhesive systems with different lowercase letters in each row. * Light-cure only control group in which core was light-cured for all groups

**Table 4 dentistry-13-00276-t004:** ANOVA table for shear bond strength.

	Mean Square	df	F	*p*
Core Material	1958.148	12	33.422	<0.001
Adhesive System	8295.646	3	141.590	<0.001
Core Material × Adhesive System	321.283	36	5.484	<0.001

**Table 5 dentistry-13-00276-t005:** Ratio of cohesive to adhesive failures in each group.

	OptiBond Universal	Scotchbond Universal Plus	Clearfil SE Bond 2	Scotchbond Universal Plus (Core Light-Cured)
Activa BioActive Restorative	3:7	4:6	8:2	7:3
Build-It FR	0:10	0:10	4:6	0:10
Bulk EZ Plus	1:9	2:8	10:0	5:5
CompCore AF	0:10	0:10	0:10	1:9
Core Paste XP Syringe	0:10	0:10	0:10	6:4
Evanesce Bulk Fill	2:8	6:4	9:1	5:5
FluoroCore 2+	0:10	0:10	1:9	8:2
Geristore	0:10	0:10	1:9	7:3
LuxaCore Z Dual	0:10	0:10	0:10	1:9
Multicore Flow	0:10	0:10	5:5	5:5
ParaCore	0:10	0:10	2:8	2:8
Rebilda DC	0:10	0:10	1:9	6:4
Filtek Bulk Fill Universal	5:5	7:3	9:1	NA

## Data Availability

The original contributions presented in this study are included in the article. Further inquiries can be directed to the corresponding author.

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
