# Peer review of "Compatibility of Dual-Cure Core Materials with Self-Etching Adhesives"

_dentistry, 2025, doi:10.3390/dj13070276_

Round 1
Reviewer 1 Report
Comments and Suggestions for Authors
This manuscript describes the shear bond strength to dentin of different core materials using three different self-etching adhesives. The scope is adequate, the methods are appropriate and comprehensive. However, the sample size calculation was not described.
The main shortcoming stems from the observe and report narrative, without much discussion of the underlying setting mechanisms of core material as well as the bonding mechanism, which is key to the success of core buildups.
The details about sectioning saw and machine should be provided.
It would be easier to follow the results if ± is inserted instead of +/-.
Author Response
Reviewer: The sample size calculation was not described.
Response:
The sample size was constrained by the availability of biologic specimens within the time window required for fresh tooth collection. To ensure bonding to freshly extracted teeth, specimens were limited to those no older than six months. Over this collection period, 260 suitable teeth were obtained and sectioned in half, resulting in 520 specimens. This allowed for a group size of 10. A post-hoc power analysis was conducted to justify this sample size, and this information has now been included in the manuscript. Pg 3 Ln 124-131
Reviewer: The main shortcoming stems from the observe and report narrative, without much discussion of the underlying setting mechanisms of core material as well as the bonding mechanism, which is key to the success of core buildups.
Response:
The manuscript has been restructured to emphasize its focus on the compatibility of various classes of dual-cure materials—specifically amine-containing, amine-free, and acid–base-containing—with different self-etch adhesives. Additionally, information about adhesive and cohesive failures has been added. Pg 3 Ln 109-111 Pg 9 Ln 272-278
Reviewer: The details about sectioning saw and machine should be provided.
Response:
Thank you. This information has now been included in the manuscript. Pg 3 Ln 133
Reviewer: It would be easier to follow the results if ± is inserted instead of +/−.
Response:
Thank you. The symbol “±” has been used throughout for consistency.
Reviewer 2 Report
Comments and Suggestions for Authors
- The study is very interesting and has high clinical relevance.
- The abstract is clear and well written.
- In the methodology section the use of shear bond testing is very good however the number of sample size that was used is 10 and that needs to be justified.
- The table 3 of the results section needs to be modified as it is not reader friendly and difficult to be understood.
- The limitation section needs to be improved focusing on the impact of thermal stress and the need for aging protocols in the future studies.
- Please be very consistent while using the chemical names through-out the manuscript.
- There are several typo errors so, please check the manuscript well.
- The reference list needs to be updated.
Overall, the manuscript is very well written and very clear. However, it needs some minor modifications before it is accepted.
Author Response
In the methodology section the use of shear bond testing is very good however the number of sample size that was used is 10 and that needs to be justified.
Response:
The sample size was constrained by the availability of biologic specimens within the time window required for fresh tooth collection. To ensure bonding to freshly extracted teeth, specimens were limited to those no older than six months. Over this collection period, 260 suitable teeth were obtained and sectioned in half, resulting in 520 specimens. This allowed for a group size of 10. A post-hoc power analysis was conducted to justify this sample size, and this information has now been included in the manuscript.
The table 3 of the results section needs to be modified as it is not reader friendly and difficult to be understood.
Response:
Thank you. The title of Table 3 has been modified to better indicate what is being presented (mean ±standard deviation) as well as the units of measurement (MPa)
The limitation section needs to be improved focusing on the impact of thermal stress and the need for aging protocols in the future studies.
Response:
Thank you. We agree that thermocycling or aging would have been ideal. This information has been added on page 10 lines 327-332.
Please be very consistent while using the chemical names through-out the manuscript.
Response:
Table 1 has been thoroughly edited to ensure consistency in the chemical names presented.
There are several typo errors so, please check the manuscript well.
Response:
The manuscript has been reviewed by all authors and checked by an online text editor to check for typos.
The reference list needs to be updated.
Response:
References 24–28, all published within the past 10 years, have been added to strengthen the currency of the citation base.
Reviewer 3 Report
Comments and Suggestions for Authors
Review of manuscript dentistry-3630960, titled: Compatibility of dual-cure core materials with self-etching adhesives.
This paper presents the results of an experimental study of the shear strength of a dentin joint with selected dual self-etch composite core materials.
The manuscript is well-organized but requires significant content revisions to enhance its readability:
Below are some of the main observations, listed chronologically by occurrence with a breakdown by chapter.
Chapter: Introduction
Develop the text with the latest publications in the field is needed. The authors used 28 literature items in the introduction, 17 of which are articles older than 10 years. Which may suggest that the introduction presents the established canons and not the current state of knowledge of the work. It is necessary to clarify the meaning of compatibility (what exactly is meant by the phrase “potential compatibility issues were noted” L:56) and emphasize the criteria for its evaluation as the subject of the work. In addition, I recommend adding a paragraph on the mechanical properties evaluated in this study, the values of the shear strength of the dentin-fill material interface obtained by other researchers should be given as a benchmark for the expected results of this work. The introduction should also describe the possible mechanisms of shear failure of the joint (e.g. cohesive or adhesive failure). What was the motivation behind this research? Why was the comparison of these materials chosen? Can subgroups be distinguished among them, e.g., based on the bonding mechanism or composite composition?
Chapter: Materials and Methods
It should be explained in detail how many teeth were used in each subgroup. According to the notation posted by the authors (13 groups, in each group different pH, 4 subgroups, 10 per group) according to my assessment does not give the given number (300?) or 10 divided 4? I suggest including a more detailed explanation. Too few samples per subgroup would lead to erroneous conclusions and would disqualify the study.
The shear test conditions should be specified, such as the area of the shear area and the range of the machine's measurement load cell.
In Table 1, I recommend adding Lot no.
Mark the reference group in Table 2. The use of bold font in table 1 is incomprehensible.
Delete line 172.
Table 3 needs to be supplemented with units.
The parameters used in Table 4 require clarification.
Figure 2, I suggest dividing, for example, into three or four, according to the value of pH adhesive materials.
The results must necessarily include a description and photos of the destruction schemes. This will complement the information on the shear strength of the joints examined in the study. This is crucial for assessing the biocompatibility of the systems used. It is necessary to develop the numerical and statistical description of the results, so I recommend moving parts of the text from the discussion section to the results section.
Chapter: Discussion
The discussion is very weak. In the discussion of results, only five literature sources were used, four of which are over ten years old. It is essential to supplement and expand this section. There is scientific literature available that includes experimental data and a literature review within this field from the past 5–10 years. This should be incorporated into the discussion of the results to enhance the scientific value and practical significance of this study.
Author Response
Reviewer: Develop the text with the latest publications in the field is needed. The authors used 28 literature items in the introduction, 17 of which are articles older than 10 years. Which may suggest that the introduction presents the established canons and not the current state of knowledge of the work.
Response:
Thank you for pointing this out. We performed updated literature searches in relevant databases and included five additional recent studies, including all references cited in Bayindir (2023). These additions contributed to a restructuring of the Introduction to better reflect the current state of knowledge.
Reviewer: It is necessary to clarify the meaning of compatibility (what exactly is meant by the phrase “potential compatibility issues were noted” L:56) and emphasize the criteria for its evaluation as the subject of the work.
Response:
Thank you for this excellent point. We have clarified that “incompatibility” refers specifically to low dentin bond strength.
Reviewer: I recommend adding a paragraph on the mechanical properties evaluated in this study the values of the shear strength of the dentin-fill material interface obtained by other researchers should be given as a benchmark for the expected results of this work.
Response:
We could only identify one other paper which also used the ISO recommended shear bond strength method to test the dentin shear bond strength with core materials. We have compared our results to the results of that study in the Discussion section.
Reviewer: The introduction should also describe the possible mechanisms of shear failure of the joint (e.g. cohesive or adhesive failure).
Response:
Thank you. The paper now describes possible failure mechanisms, including adhesive and cohesive failure, to better contextualize shear strength results.
Reviewer: What was the motivation behind this research? Why was the comparison of these materials chosen? Can subgroups be distinguished among them, e.g., based on the bonding mechanism or composite composition?
Response:
The purpose of the study was to evaluate whether any dual-cure core materials are compatible with self-etch adhesives. Materials were selected to represent a range of chemistries—amine-containing, amine-free, and acid–base-containing—to enable a comparative analysis. This rationale has been added to the Introduction.
Reviewer: It should be explained in detail how many teeth were used in each subgroup. According to the notation posted by the authors (13 groups, in each group different pH, 4 subgroups, 10 per group) according to my assessment does not give the given number (300?) or 10 divided 4? I suggest including a more detailed explanation. Too few samples per subgroup would lead to erroneous conclusions and would disqualify the study.
Response:
Clarification has been added regarding sample numbers. Out of 300 teeth collected, 260 met inclusion criteria and were sectioned in half to create 520 specimens. A post-hoc power analysis has also been included to support the sample size.
Reviewer: The shear test conditions should be specified, such as the area of the shear area and the range of the machine's measurement load cell.
Response:
This information has now been added to the Methods section, including the shear test area and the range of the testing machine’s load cell.
Reviewer: In Table 1, I recommend adding Lot no.
Response:
This has been removed as suggested.
Reviewer: Mark the reference group in Table 2. The use of bold font in table 1 is incomprehensible.
Response:
The reference group has been clearly indicated in Table 2, and bold font in Table 1 has been removed for consistency.
Reviewer: Delete line 172.
Response:
This line has been deleted.
Reviewer: Table 3 needs to be supplemented with units.
Response:
Thank you for noting this. Units are now included in the title of Table 3.
Reviewer: The parameters used in Table 4 require clarification.
Response:
Thank you. This has been corrected to properly label the table as an ANOVA table.
Reviewer: Figure 2, I suggest dividing, for example, into three or four, according to the value of pH adhesive materials.
Response:
We appreciate this suggestion. However, we prefer to present Figure 2 (now Figure 3) as a combined figure to facilitate comparison across all adhesives and core materials. This format allows readers to more easily visualize the influence of adhesive type on each material and vice versa.
Reviewer: The results must necessarily include a description and photos of the destruction schemes. This will complement the information on the shear strength of the joints examined in the study. This is crucial for assessing the biocompatibility of the systems used. It is necessary to develop the numerical and statistical description of the results, so I recommend moving parts of the text from the discussion section to the results section.
Response:
Thank you for this suggestion. We have conducted a failure mode analysis and included representative images of each failure type. Table 5 has been added to categorize the failure modes, and a corresponding description has been moved to the Results section to strengthen that part of the manuscript.
Reviewer: The discussion is very weak. In the discussion of results, only five literature sources were used, four of which are over ten years old. It is essential to supplement and expand this section. There is scientific literature available that includes experimental data and a literature review within this field from the past 5–10 years. This should be incorporated into the discussion of the results to enhance the scientific value and practical significance of this study.
Response:
Thank you for this valuable feedback. We have significantly expanded the Discussion section, including recent studies published within the past 5–10 years, to enhance the scientific depth and relevance of the findings.
Round 2
Reviewer 3 Report
Comments and Suggestions for Authors
Thank you for considering my comments. The necessary explanations have been added, and the manuscript has been significantly improved. The Introduction chapter has been developed. The Materials and Methods section is appropriately structured and sufficient to assess the assumptions underlying the work's purpose. The results are presented clearly and in an understandable manner. The Discussion section has been substantially expanded and incorporates recent studies. I believe readers will appreciate the effort invested in developing the article.
Author Response
Thank you again for your thoughtful and constructive feedback throughout the review process. Since no further changes have been requested at this stage, we appreciate your time and consideration in helping us improve our manuscript.